# Preparation of Copper Ion Adsorbed Modified Montmorillonite/Cellulose Acetate Porous Composite Fiber Membrane by Centrifugal Spinning

**DOI:** 10.3390/polym14245458

**Published:** 2022-12-13

**Authors:** Hongjing Zhang, Qingyuan Mu, Xiaotian Yu, Ke Zhou, Xu Chen, Haitao Hao, Yongqiang Li

**Affiliations:** 1Key Laboratory of Advanced Textile Materials and Manufacturing Technology, International Silk Institute, College of Textiles, Ministry of Education, Zhejiang Sci-Tech University, Hangzhou 310018, China; 2Engineering Research Center for Eco-Dyeing and Finishing of Textiles, Ministry of Education, Zhejiang Sci-Tech University, Hangzhou 310018, China; 3Key Laboratory of Intelligent Textile and Flexible Interconnection, Ministry of Education, Zhejiang Sci-Tech University, Hangzhou 310018, China; 4Tongxiang Research Institute, Zheijiang Sci-Tech University, Tongxiang 345000, China

**Keywords:** cellulose acetate, montmorillonite modification, porous fiber membrane, adsorption of heavy metal ions, adsorption kinetic

## Abstract

The natural adsorption material montmorillonite (MMT) was selected, and cellulose acetate (CA) was used as the loading substrate to design and prepare a kind of green and environment-friendly recyclable porous composite fiber membrane with good heavy metal ion adsorption performance. Acetic acid modified montmorillonite (HCl-MMT), sodium dodecyl sulfonate modified montmorillonite (SDS-MMT), and chitosan modified montmorillonite (CTS-MMT) were prepared by inorganic modification and organic modification, and the porous MMT/CA composite fiber membrane was constructed by centrifugal spinning equipment. The morphological and structural changes of MMT before and after modification and their effects on porous composite fiber membranes were investigated. The morphology, structure, and adsorption properties of the composite fibers were characterized by scanning electron microscopy (SEM) and atomic absorption spectrometry (ASS). The experimental results showed that the maximum adsorption capacity of Cu^2+^ on the prepared 5 wt% CTS-MMT composite fiber membrane was 60.272 mg/g after 10 h static adsorption. The adsorption of Cu^2+^ by a porous composite fiber membrane conforms to the quasi-second-order kinetic model and Langmuir isothermal adsorption model. The main factor of the Cu^2+^ adsorption rate is chemical adsorption, and the adsorption mechanism is mainly monolayer adsorption.

## 1. Introduction

Heavy metal ions are one of the main sources of water pollution [1]. Most heavy metal ions are carcinogens, posing a threat to ecological balance and human health [2]. Cu^2+^ is an essential trace element for the human body. However, too high a concentration of copper will lead to serious health problems such as brain and kidney issues and anemia [3]. The discharge of heavy-metal-polluted wastewater (e.g., brine) degrades water quality, and thus water cannot be directly used for potable water (via desalination) and industrial applications [4,5,6]. Therefore, it is necessary to remove residual copper ions in water. Many articles have reported various sewage treatment methods containing heavy metal ions, including adsorption, ion exchange, chemical precipitation, membrane filtration, electrochemical technology, and reverse osmosis [7,8,9,10]. As a water treatment technology with simple operation, obvious purification effects, large treatment capacity, and low cost, the adsorption method has been widely used in the treatment of heavy metal pollution in recent years [11,12,13]. Anirudhan et al. [14,15] prepared a Polyitaconic acid/methacrylic acid grafted nano cellulose/bentonite composite [P(IA/MAA)g-NC/NB] with multi-carboxyl functional groups through free radical polymerization. The material has good adsorption capacity for Co^2+^ in nuclear power plant wastewater and Th^4+^ in seawater and groundwater. The adsorption mechanism study shows that the adsorption process is mainly ion exchange complexation. Pingxiao Wu et al. [16] prepared hydroxyl iron pillared bentonite through modification and studied its adsorption of cadmium, with an adsorption capacity of 25.7 mg/g. Shahid et al. [17] prepared poly (N-isopropylacrylamide acrylamide methacrylic acid) [p(NAM)] micro gel by free radical precipitation polymerization, which has good adsorption capacity for cobalt ions in water. Arif et al. [18] reviewed the application of various gold nanoparticle supported micro gels to treat metal ions. The adsorption materials prepared by the above synthetic methods have the problems of complex process and difficult biodegradation. Therefore, it is necessary to select green natural adsorption materials to treat metal ions.

Montmorillonite has a layered structure, large specific surface area, exchangeable interlayer cations, and other properties, which makes it have significant advantages in eliminating heavy metal ion pollution [19,20,21]. However, due to the existence and strong suspension of impurities such as oxides, their adsorption capacity is reduced and it is easy to cause secondary pollution [22,23]. To make montmorillonite more widely applicable, it needs to be modified and loaded on the carrier [24]. At present, the modification methods include acid activation modification [25], organic modification [26], and inorganic–organic composite modification [27]. Yonggui Chen et al. [28] prepared sodium polyacrylate bentonite (SPB) with low permeability and chemical resistance through modification experiments. The adsorption results show that the unit adsorption effect of SPB on Pb under acidic conditions is obvious, and the maximum unit adsorption amount reaches 72.89 mmol/100 g, which is nearly 20% higher than that of natural bentonite. Pawar et al. [29] activated the bentonite with sulfuric acid, and the specific surface area of the treated bentonite increased by 3.3 times and the pore volume increased by 2.75 times. The adsorption of Pb^2+^ and Cu^2+^ are 21.359 mg/g and 9.793 mg/g, respectively, so acidified bentonite can be used to treat water-containing wastes polluted by Pb (II) and Cu (II). Li et al. [30] prepared TiO_2_ pillared bentonite (TiO_2_/MMT) as the adsorption material to treat As (V) and As (III) in solution. As a result, the adsorption capacity increased rapidly in the first 20 min, and the adsorption reached equilibrium within 1 h. Therefore, TiO_2_/MMT as an adsorption material showed a good adsorption effect on arsenic. Ren et al. [31] modified magnetic bentonite with a cationic surfactant, amphoteric surfactant, and anionic surfactant to remove cadmium (Cd^2+^). The results showed that the ratio of BS-MBt to other adsorbents reached the highest Cd^2+^ adsorption capacity (233.19 mmol/kg). However, most MMTs and modified MMTs are directly put into wastewater treatment in the form of powder during use. Due to their high suspension in water, they are not easy to precipitate after adsorption, which makes it difficult to separate solids from the liquid and recover them [32,33]. Therefore, it is necessary to prepare composite materials with the help of the supporting substrate to solve the problem of difficult separation after adsorption. Porous fibrous materials have a high specific surface area and good physical and chemical properties, which can not only make up for the shortcomings of MMT powder but also obtain a more excellent adsorption effect through the synergistic effect with the loaded substrate.

Cellulose acetate (CA) is an ideal loading substrate with biodegradability, light transmittance, and a certain hardness. It is considered to be an excellent candidate material for the preparation of bio-based polymer/clay nanocomposites [34]. At present, the CA micro/nanofiber membrane is usually prepared by electrospinning technology, which has the characteristics of a large specific surface area and high porosity and can be applied to adsorption materials. However, the output of nanofiber prepared by this method is very low, which makes it difficult to carry out industrialized production and means it cannot meet the needs of society [35]. Centrifugal spinning has high production efficiency, a wide range of raw materials, and a controllable fluffy degree of fiber aggregate. It is a preparation method for micro/nanofiber with industrial application prospects [36,37].

To solve the above problems, three kinds of modified MMT were prepared by two modification methods and loaded on CA fiber to prepare an MMT/CA porous composite micro nanofiber membrane. The influence of MMT obtained by different modification methods on the composite fiber membrane was studied, and the influence of the amount of modified MMT on the morphology and properties of the porous composite fiber membrane was proved. The centrifugal spinning controllable preparation of a porous composite fiber membrane was realized, and it was applied to the adsorption of heavy metal ions in water.

## 2. Experimental Methods

### 2.1. Materials and Devices

The following reagents were used in the experiment: cellulose acetate (CA, mw = 100,000, American Acros), dimethyl sulfoxide (DMSO, analytically pure, Tianjin Kemio Chemical Reagent Co., Ltd.), montmorillonite (MMT, 1500 mesh, product of Henan Xinyang Moon Fruit Material Factory), dichloromethane (DCM, analytical reagent, Hangzhou Sanying Chemical Reagent Co., Ltd.), hydrochloric acid (HCl, analytically pure, Hangzhou Gaojing Fine Chemical Co., Ltd.), chitosan (CTS, analytically pure, Zhejiang Golden Shell Biochemical Co., Ltd.), Sodium dodecyl sulfonate (SDS, analytically pure, Hefei BASF Biotechnology Co., Ltd.), glacial acetic acid (analytical reagent, Hangzhou Sanying Chemical Reagent Co., Ltd.), copper sulfate (CuSO_4_·5H_2_O, analytically pure, Guangdong Guanghua Technology Co., Ltd.), and disodium ethylenediaminetetraacetate (EDTA-2Na, analytical pure, Hangzhou Pharmaceutical Chemical Reagent Factory).

The following equipment was used in the experiment: Pl203 electronic balance (Guangzhou Instrumental Laboratory Technology Co., Ltd.), IKA RET basic heating magnetic stirrer (Guangzhou Yike Laboratory Technology Co. Ltd.), vacuum over of-6020 vacuum drying oven (Shanghai Heheng Instrument Equipment Co. Ltd.), and pH-220 pen pH meter (Hangzhou Wenwei Instrument Co. Ltd.).

### 2.2. Experimental Process

#### 2.2.1. Preparation of Modified MMT

(a) HCl solution with a mass fraction of 36.5% was added to ultra-pure water and diluted to 150 mL HCl dilute solution with a mass fraction of 2%. An amount of 3.0 g montmorillonite was accurately weighed and added to the prepared HCl diluent. The constant temperature magnetic stirring water bath pot was used, the heating temperature was 50 °C, the rotation speed was set to 600 rpm, and the stirring was continued for 5 h under this condition. The mixed solution after stirring was centrifuged with a high-speed centrifuge (8000 rpm), the supernatant after centrifugation was discarded, the centrifuged precipitate was washed with purified water to remove HCl remaining in MMT, and the washing was repeated several times until the pH of MMT eluate reached 6. Centrifuging was performed again, and then the centrifuged products were dried in a vacuum drying oven with a constant temperature of 80 °C. Following this, the dried products were fully ground, and finally the acid-activated montmorillonite was prepared.

(b) The anionic surfactant SDS was accurately weighed and dissolved in ultra-pure water, and 150 mL of 0.1 mol/L SDS solution was prepared. Then, 3.0 g MMT was dispersed in the prepared SDS solution in the ratio of 1:60 (*m*/*v*), and the MMT/SDS mixed system was mixed and reacted at 80 °C and 600 rpm for 8 h by magnetic stirring. The reaction mixture was separated with a high-speed centrifuge (8000 rpm), and the precipitate was washed several times with 50% ethanol and centrifuged. Finally, the remaining solid was dried at 80 °C, and the dried product was fully ground to prepare SDS-MMT.

(c) A certain amount of glacial acetic acid (99%) was measured and added to ultra-pure water to prepare a 200 mL acetic acid aqueous solution with a mass fraction of 5%. It was stirred evenly and set aside. Then, 2 g CTS was accurately weighed and it was added to the prepared 5% acetic acid aqueous solution. It was stirred in a 60 °C water bath for 0.5 h to completely dissolve it to form a 10 g/L CTS solution. Then, 2 g MMT was accurately weighed, and it was added to the prepared CTS solution. The above mixture was placed on the collector-type constant temperature heating magnetic stirrer and stirred for 12 h at 35 °C and 600 rpm. The reaction mixture was separated with a high-speed centrifuge (8000 rpm) and the supernatant was discarded. The separated precipitated products were placed in a vacuum drying oven (80 °C) for drying, and then fully ground to finally prepare CTS-MMT.

#### 2.2.2. Preparation of MMT/CA Porous Composite Fiber Membrane

Certain amounts of CA and MMT were weighed with an electronic balance, and they were then dissolved in the mixed solvent of DCM/DMSO (8:2 *m*/*m*) and placed in the sample bottle. The sample bottle was sealed with fresh-keeping film, raw material tape, and sealing film. After stirring for 12 h at 40 °C with a magnetic stirrer until it was completely dissolved, a centrifugal spinning solution with uniform MMT/CA dispersion was finally prepared. The centrifugal spinning machine was used with the following settings: the rotation speed of the centrifuge was set to 8000 r/min, the nozzle aperture to 0.4 mm, and the distance from the spinneret hole to the collecting rod to 12.0 cm. An amount of 5 mL of spinning solution and spinning head was taken, and it was stretched by the centrifugal force of the centrifuge and the solvent was volatilized rapidly. Finally, the porous composite fiber membrane was prepared.

#### 2.2.3. Copper Ion Adsorption Experiment

An amount of 100 mg of prepared MMT/CA porous composite fiber membrane was immersed in a solution containing Cu^2+^ (100 mg/L). The solution was placed in a shaking water bath for shaking, allowing the fiber membrane to fully contact the heavy metal ion solution. The fiber automatically adsorbed at a constant temperature of 25 °C, and the adsorption time was 24 h. After adsorption, the composite fiber membrane was taken out, soaked in ultra-pure water for 2 h, and dried for standby.

The adsorbed solution was diluted and the Cu^2+^ concentration in the solution was detected by atomic absorption spectrometer. Then, the Cu^2+^ adsorption capacity *q*_e_ of the porous composite fiber membrane was calculated according to Formula (1).
(1)$$q_{e}=\frac{\left(C_{0}-C_{e}\right)\cdot V}{m}$$

In the formula, *q_e_* is the equilibrium adsorption amount of Cu^2+^ after reaching the adsorption equilibrium of fiber membrane, mg/g; *c*_0_ is the initial concentration of Cu^2+^ in the solution prepared at the beginning of the experiment before adsorption, mg/L; *c_e_* is the concentration of Cu^2+^ in the solution, mg/L, after reaching the adsorption equilibrium of the fiber membrane; *m* is the mass of the porous composite fiber membrane in the adsorption experiment, mg; and *V* is the volume of the Cu^2+^ solution configured in the experiment, L.

### 2.3. Instruments and Characterization

The surface morphology of the composite fiber film was observed by scanning electron microscope (SEM, Germany, Carl Zeiss SMT Pte Ltd.), and the particle size distribution of the fiber was analyzed. Fourier transform infrared spectroscopy (FTIR, German Brooke spectrometer company, the number of scans was 32) was used to analyze the chemical structure of the samples by the potassium bromide mixed compression method. Analysis of the element composition of the composite fiber was conducted by X-ray photoelectron spectroscopy (K-Alpha, Thermo Scientific). The material composition and crystal structure of the composite fiber were analyzed by X-ray diffraction (XRD, ARL XTRA, Thermo ARL). The concentration change in Cu^2+^ in the solution was detected by an AA-110 atomic absorption spectrometer (ASS, Beijing Purcell General Co., Ltd.).

## 3. Results and Discussion

### 3.1. Effect of Different Mass Fractions of MMT on Porous Composite Fiber Membrane

The porous composite fiber membranes with a different mass fraction of MMT (0~3%) and CA were prepared under the same spinning parameters. The influence of MMT mass fraction on the morphology of porous composite fibers was analyzed, and the relationship between MMT mass fraction and fiber diameter was investigated. Figure 1 shows SEM images and fiber diameter distribution of porous composite fiber membranes loaded with different mass fractions of MMT. According to the fiber diameter test results, the average fiber diameter corresponding to composite fiber membranes with MMT contents of 0%, 1%, 2%, and 3% is 11.74 μm, 12.04 μm, 13.49 μm, and 15.01 μm. With the increase in MMT content, the fiber surface still showed a porous structure, but the average diameter of the porous composite fiber increased gradually. This may be due to the higher mass fraction of MMT and the higher jet mass per unit spinning solution. Therefore, under the same centrifugal force, the spinning liquid jet is not easy to stretch, so the diameter of composite fibers increases with the increase in MMT content. When the addition of MMT is more than 3%, the porous CA fiber skeleton in the system cannot support too much MMT, which leads to difficulties in filamentation, and the fibers are mostly broken.

### 3.2. SEM Analysis of Modified MMT

SEM images of various MMTs before and after modification are shown in Figure 2. Figure 2a shows that unmodified MMT mainly exists in the form of larger particles and has a layered structure. Figure 2b shows hydrochloric-acid-modified MMT (HCl-MMT). The modified MMT has a relatively small particle size, high dispersion, and large specific surface area. It is mainly because H^+^ in a hydrochloric acid solution can dissolve some acid-soluble metal oxides and impurities in MMT, so that the lamellar structure of MMT becomes loose and the agglomerated particles are dispersed. The particle size of SDS-modified MMT (Figure 2c) decreases. As SDS enters the MMT lamellar structure, the agglomerated particles become dispersed, some particle multilayer structures are spread in the form of single layers, and the specific surface area of MMT increases. Figure 2d shows chitosan-modified MMT (CTS-MMT). Compared with the morphology before modification, it can be seen that the surface of CTS-MMT is covered by a layer of flakes, and the specific surface area has increased.

### 3.3. Surface Analysis of MMT/CA Porous Composite Fiber Membrane

Figure 3 shows the SEM images and corresponding fiber diameter distribution of composite fiber membranes prepared by three modified MMTs at different concentrations. It can be seen from Figure 3 that the modified MMT can prepare composite fiber membranes with porous surface structures. Under the same MMT addition (3 wt%), the average diameter of the prepared HCl-MMT/CA porous composite fiber membrane fiber is larger (average diameter is 15.48 μm), the spinning effect is relatively poor, and some fibers are twisted. However, SDS-MMT/CA and CTS-MMT/CA porous composite fiber membranes have better fiber morphology and smaller average fiber diameter (average diameters are 14.82 μm and 14.06 μm, respectively). This may be due to the existence of a long-chain molecular structure in SDS-MMT and CTS-MMT, which can be fully entangled with the CA molecular chain in the process of spinning solution preparation and stirring and can be more evenly distributed in the spinning solution system. The spinning effect of the prepared composite fiber is better. Due to the small particle size of HCl-MMT, it is easy to agglomerate in the spinning process, resulting in a poor spinning effect and composite fiber morphology. When the addition amount is 4 wt%, because the agglomeration phenomenon of HCl-MMT is intensified in the spinning process, the spinning solution cannot form a continuous jet, so the composite fiber membrane cannot be prepared. For SDS-MMT, due to the limited load of CA fibers, too much SDS-MMT will partially agglomerate. Although the composite fiber membrane is successfully prepared, its fiber morphology is poor, and the direct distribution of fibers is uneven (average diameter of 15.78 μm). CTS-MMT has a certain viscosity due to the existence of chitosan molecules on its surface, which can provide CA fibers with certain bonding properties during the spinning process so that its jet is not easily interrupted. Therefore, the morphology of the prepared composite fiber membrane is still good, and the surface roughness changes. However, when the addition of CTS-MMT reaches 5 wt%, the diameter of the composite fiber membrane fiber is relatively thick, the surface roughness of the fiber further increases, and the porous structure of the fiber surface begins to disappear gradually. When the addition amount exceeds 5 wt%, the maximum load of CA fiber has been exceeded, and the porous composite fiber membrane with the corresponding addition amount cannot be successfully prepared.

In order to further study the distribution of MMT on fibers, TEM images of samples were observed. As shown in Figure 4, compared with the pure CA fiber membrane, with the increase in the amount of modified MMT added, its distribution on the fiber surface also increases. The existence of MMT can be clearly observed on the porous fiber membrane prepared by the three modified MMTs.

Figure 5 shows the SEM diagram of the intercepted part of the high-load-modified MMT composite fiber membrane. It can be seen from the figure that when the addition amount of HCl-MMT is 4 wt%, the spun fiber is flat, and some fibers are melted and agglomerated together. When the addition amount of SDS-MMT is 5 wt%, the spun fiber is twisted and bonded and the amount is small. When the addition amount of CTS-MMT is 6 wt%, the spun fibers are different in thickness and twisted together. Therefore, the maximum spinnable addition amounts of each modified MMT are 3 wt% (HCl-MMT), 4 wt% (SDS-MMT), and 5 wt% (CTS-MMT), respectively.

### 3.4. Infrared Spectrum Analysis of Modified MMT

Figure 6 shows the FTIR spectrum of MMT and modified MMT. It can be seen from Figure 6 that the peak at 3416 cm^−1^ in the four curves belongs to the -OH stretching vibration peak in MMT, indicating that the basic structure of the modified MMT is not damaged. New absorption peaks appear in the spectral curves c and d of SDS-MMT and CTS-MMT, while no new absorption peaks appear in the spectral curve b of HCl-MMT. The absorption peaks of curve c at wavelengths 2924 cm^−1^ and 2852 cm^−1^ correspond to the characteristic absorption peaks of -CH_3_ and -CH_2_ in SDS, respectively [38], which proves that SDS functional groups are successfully embedded in MMT. The new absorption peak at the wavelength of 1563 cm^−1^ in curve d belongs to the characteristic absorption peak of the amino group in CTS [32]. Due to the small concentration of CTS selected in the modification process, the characteristic absorption peak of the amino group at the wavelength of 1563 cm^−1^ in the measured data is small. Due to the dissolution of some impurities in the natural MMT, the measured infrared curve b of HCl-modified MMT is not significantly different from the natural MMT curve a. The feasibility of several modification methods was proved by FTIR analysis of MMT.

### 3.5. EDS Analysis of Porous Composite Fiber Membrane

Figure 7 shows the EDS diagram of the composite fiber membrane and the EDS diagram of CTS prepared by the three modified MMTs at their respective maximum addition concentrations. According to the test results, the additional amount of MMT in the HCl-MMT composite fiber membrane is the same as that in the natural MMT composite fiber membrane (3 wt%). However, comparing Figure 7b with the EDS of the natural MMT composite fiber membrane (Figure 7a), it can be seen that the proportions of O, Al, and Si in the EDS diagram of HCl-MMT composite fiber membrane increased, which may be due to the more uniform dispersion of MMT modified by hydrochloric acid, which can obtain a larger load in spinning. SDS-MMT and CTS-MMT composite fiber films, due to their large amount of self-addition and easier preparation, account for a relatively large proportion of Al and Si elements in their EDS diagrams. According to Figure 7c, CTS molecules mainly contain C, N, and O elements, and the relative content of the N element is relatively small. A small amount of N-element peak can be observed in Figure 7d, indicating that CTS-MMT has a large content in the composite fiber membrane.

### 3.6. XRD Analysis of Porous Composite Fiber Membrane

To study the change in the layered structure of the modified MMT, X-ray diffraction (XRD) was used to characterize it. Figure 8 shows the XRD test results of MMT before and after modification. XRD test results combined with Bragg’s law 2dsin θ = λ (Cu target, wavelength λ 0.15406 nm) can calculate the change in MMT layer spacing before and after modification. Compared with the natural MMT before modification, the diffraction angle of the characteristic crystal plane (001) of the three modified MMTs has shifted. It can be seen from the formula that the layer spacing does increase gradually when the diffraction angle shifts to a small angle. The characteristic diffraction peak of natural MMT is at 2θ = 9.18°, corresponding layer spacing d = 0.96257 nm. The shift of the HCl-MMT characteristic diffraction peak is small at 2θ = 9.06°, with a corresponding layer spacing d = 0.97530 nm. The characteristic diffraction peak of CTS-MMT is at 2θ = 8.62°, with layer spacing d = 1.02498 nm. The maximum shift of the SDS-MMT characteristic diffraction peak is 2θ = 8.20°, with a layer spacing d = 1.07738 nm. According to the calculation results, the interlayer spacing of HCl-modified MMT increased by 0.01273 nm due to the removal of interlayer impurities, while the interlayer spacing of CTS-modified MMT and SDS-modified MMT increased by 0.06241 nm and 0.11481 nm, respectively, due to CTS and SDS entering into the interlayer of MMT during the modification process. The shift of the MMT characteristic diffraction peak and the change in layer spacing before and after modification further illustrate the feasibility of the modification method.

### 3.7. Kinetics of Adsorption of Cu^2+^ on Porous Composite Fiber Membrane

Before the experiment, the influence of the pH value of the solution on the adsorption performance of the fiber membrane was tested. When the pH value is less than 5, the -NH_2_ in the CTS easily forms -NH_3_^+^ with the H^+^ in the solution under acidic conditions, thus repelling Cu^2+^. When the pH value is greater than 5, with the decrease in the concentration of H^+^ in the solution, Cu^2+^ in the solution is easy to ionize in the water, forming hydrated ions, making it difficult to be adsorbed. When pH = 5, the porous composite fiber membrane had the best adsorption effect on Cu^2+^. The porous composite fiber membrane was added to the Cu^2+^ adsorption solution (100 mg/L) with a pH value of 5 for the adsorption experiment. A certain amount of sample solution was extracted from the adsorption solution at 0.5, 1, 2, 4, 8, 12, and 24 h, respectively, for the atomic absorption spectrometry test. The relationship curve between adsorption time and adsorption capacity is shown in Figure 9. Due to a large number of binding points on the porous composite fiber and the high concentration of Cu^2+^ in the solution, the adsorption capacity increases almost linearly [39], and the growth rate of the CTS-MMT porous composite fiber membrane is the largest. With the extension of adsorption time, due to the decrease in Cu^2+^ concentration in the solution and the gradual reduction in binding points, the adsorption speed slows down. When the adsorption reaches a certain time, the adsorption capacity of the porous composite fiber membrane for Cu^2+^ reaches saturation. The adsorption capacity of the 3% MMT composite fiber membrane reached the maximum value (44.243 mg/g) in 12 h, the 3% HCl-MMT composite fiber membrane reached the maximum adsorption capacity (46.155 mg/g) in 10 h, and the 4% SDS-MMT and 5% CTS-MMT composite fiber membranes reached the maximum adsorption capacity (52.381 mg/g and 60.272 mg/g) in 8 h. The reason is that the maximum spinning addition amount of CTS-MMT is 5 wt%. Compared with the other two modified MMTs, it can obtain a larger spinning addition amount, and the adsorption effect of heavy metal ions is more obvious.

To further analyze the adsorption performance of porous composite fiber membranes, the test data were fitted with a kinetic model. The quasi-first-order dynamics and quasi-second-order dynamics models were used to fit and analyze them, respectively. The equation of the quasi-first-order dynamics model is shown in Formula (2):(2)$$\log\left(q_{e}-q_{t}\right)=\log q_{e}-\frac{K_{1}}{2.303}t$$

In the formula, *q_e_* is the adsorption amount when the adsorption reaches equilibrium, mg/g, *q_t_* is the Cu^2+^ adsorption at time t, mg/g; *t* is the adsorption time, h; and *K*_1_ is the quasi-first-order adsorption rate constant, h^−1^.

The equation of the quasi-second-order dynamics model is shown in Formula (3):(3)$$\frac{t}{q_{t}}=\frac{1}{K_{2}q_{e}^{2}}+\frac{t}{q_{e}}$$

In the formula, *q_e_*, *q_t_*_,_ and *t* are the same as in the quasi-first-order kinetic model, and *K*_2_ represents the quasi-second-order adsorption rate constant, g/mg·h^−1^.

According to the fitting results of Cu^2+^ adsorption kinetics of the 5 wt% CTS-MMT composite fiber membrane (Figure 10) and fitting parameters (Table 1), the correlation coefficient of the quasi-first-order kinetic model is 0.885, and the quasi-second-order correlation coefficient is 0.995. The saturated adsorption capacity obtained from the quasi-second-order kinetic model is 65.92 mg/g, and the actual measured value is 60.272 mg/g. The equilibrium adsorption capacity fitted by the quasi-second-order kinetic model is closer to the actual adsorption capacity of a porous composite fiber membrane. Therefore, the adsorption process is more suitable to be explained by the quasi-second-order kinetic model; that is, the main factor affecting the adsorption rate of the porous composite fiber membrane is chemical adsorption, and the adsorption process is coordinated by a variety of kinetic mechanisms to achieve the maximum adsorption capacity.

### 3.8. Study on Adsorption Isotherm of Cu^2+^ on Porous Composite Fiber Membrane

Figure 11 is the adsorption isotherm of Cu^2+^ on a 5 wt% CTS-MMT composite fiber membrane. It can be seen from the test results in the figure that with the increase in the initial concentration of Cu^2+^ in the adsorption solution, the adsorption capacity of the porous composite fiber also increases. When the concentration of Cu^2+^ increases to 100 mg/L, its adsorption capacity will no longer continue to increase with the increase in Cu^2+^ concentration, indicating that at this concentration, the adsorption of Cu^2+^ by the porous composite fiber membrane has reached adsorption equilibrium, and the maximum adsorption capacity of Cu^2+^ is 60.272 mg/g. The main reason for this is that when the quality of the composite fiber membrane is certain, with the continuous increase in Cu^2+^ concentration in the adsorption solution, its mass transfer promotion ability also increases. Cu^2+^ is easier to move to the surface of the porous composite fiber membrane, making the adsorption easier, and increasing the adsorption capacity. However, the number of active sites on the composite fiber membrane is certain. When it is filled with Cu^2+^, the maximum adsorption capacity of the composite fiber membrane will not change, which can be considered adsorption saturation.

The equilibrium adsorption mechanism was explored between the porous composite fiber membrane and Cu^2+^ solution, and Langmuir and Freundlich adsorption isotherm models were used to linearly fit its adsorption isotherm. The linear equations of the two adsorption isotherm models can be described by Formulas (4) and (5):(4)$$\frac{C_{e}}{q_{e}}=\frac{C_{e}}{q_{m}}+\frac{1}{bq_{m}}$$
(5)$$\ln q_{e}=\ln K_{F}+\frac{\ln C_{e}}{n}$$

In the formula, *C_e_* is the concentration of Cu^2+^ in the solution at adsorption equilibrium, mg/L; *q_e_* is the adsorption amount of Cu^2+^ when the adsorption reaches equilibrium, mg/g; *q_m_* is the maximum adsorption capacity of Cu^2+^ when the adsorption reaches saturation, mg/g; *b* is the Langmuir constant, L/mg; *K_F_* is the Freundlich constant, L/mg; and *n* is the Freundlich constant.

The fitting results of the isothermal adsorption model are shown in Figure 12 and Table 2. The fitting correlation coefficients R^2^ of the two isothermal adsorption models are 0.993 (Langmuir model) and 0.889 (Freundlich model), respectively. The Langmuir adsorption isotherm is based on the assumption that the adsorbent surface is uniform, and the adsorption is regular when the adsorption reaches equilibrium in the case of the single molecular layer with the saturated adsorption capacity value. The Freundlich adsorption isotherm belongs to real adsorption, and there is no saturated adsorption capacity value. Therefore, the adsorption process of the 5 wt% CTS-MMT composite fiber membrane for Cu^2+^ is more suitable to be described by the Langmuir adsorption isotherm model, that is, the adsorption process is mainly monolayer adsorption, and each of the adsorption sites on the composite fiber membrane have the same activity and do not affect each other [40]. The maximum equilibrium adsorption capacity fitted by this model is 64.023 mg/g, which is larger than the actual maximum adsorption capacity of 60.272 mg/g, indicating that the prepared porous composite fiber membrane has great potential for Cu^2+^ adsorption. According to the empirical parameter 1/n (0.298) in the Freundlich adsorption model, which is between 0–1, it shows that the porous composite fiber membrane maintains positive adsorption during the adsorption of Cu^2+^.

### 3.9. Study on the Recycling of Porous Composite Fiber Membrane

The porous composite fiber membrane of adsorbed Cu^2+^ was desorbed with an EDTA-2Na solution (100 mg/L). It was soaked in the prepared desorption solution (25 °C, 2 h), then soaked in ultrapure water, rinsed several times, dried naturally after desorption, and the vacuum dried. After a period of time, the next adsorption desorption cycle was carried out for the Cu^2+^ solution with the same concentration, and the concentration of Cu^2+^ in the desorption solution was determined by atomic absorption spectrometer. The re-adsorption efficiency (*Re*) is calculated by Formula (6):(6)$$R_{e}=\frac{q_{d}}{q_{1}}\times 100\%$$

In the formula:

*q*_1_: The first Cu^2+^ equilibrium adsorption capacity of the porous composite fiber membrane, mg/g;

*q_d_*: Cu^2+^ equilibrium adsorption capacity of the porous composite fiber membrane after desorption, mg/g.

It can be seen from Figure 13 that after five desorption cycles, the 3 wt% natural MMT composite fiber membrane and the 5 wt% CTS-MMT composite fiber membrane have certain re-adsorption capacity. Among them, the chitosan-modified MMT composite fiber membrane has a relatively high re-adsorption effect (83%) after five desorption cycles. In this experiment, the regeneration of the CTS-MMT composite fiber membrane with strong adsorption capacity was also explored. The composite fiber membrane after the desorption cycle was dried, weighed, and prepared into a spinning solution for secondary spinning. As shown in Figure 14, by comparing the SEM images of the CTS-MMT composite fiber membrane prepared by two spinning processes, it can be seen that the surface of the composite fiber membrane prepared by re-dissolution and re-spinning still has a porous structure, and can maintain a good fiber morphology, which further illustrates that the selected load substrate CA has a certain renewable recycling usability.

## 4. Conclusions

In this paper, MMT was modified by organic and inorganic methods. The porous MMT/CA composite fiber membrane with porous structure and good Cu^2+^ adsorption properties was successfully prepared by mixing MMT and CA and adopting efficient spinning and centrifugal spinning technology. The prepared porous composite fiber membrane was applied to the adsorption of Cu^2+^. When pH = 5, the maximum adsorption capacity of the 5 wt% CTS-MMT composite fiber membrane for Cu^2+^ could reach 60.272 mg/g after 10 h static adsorption. The fitting analysis of the data of Cu^2+^ adsorption equilibrium shows that the adsorption process is more suitable to be described by the quasi-second-order kinetic model and Langmuir isothermal adsorption model. The prepared porous composite fiber membrane can still retain an 83% Cu^2+^ adsorption effect after five desorption cycles. The composite fiber membrane with porous structure can be prepared again by dissolving and spinning the composite fiber membrane after desorption, which shows that the selected load base material, CA, has a certain recyclability.

## Figures and Tables

**Figure 1 polymers-14-05458-f001:**
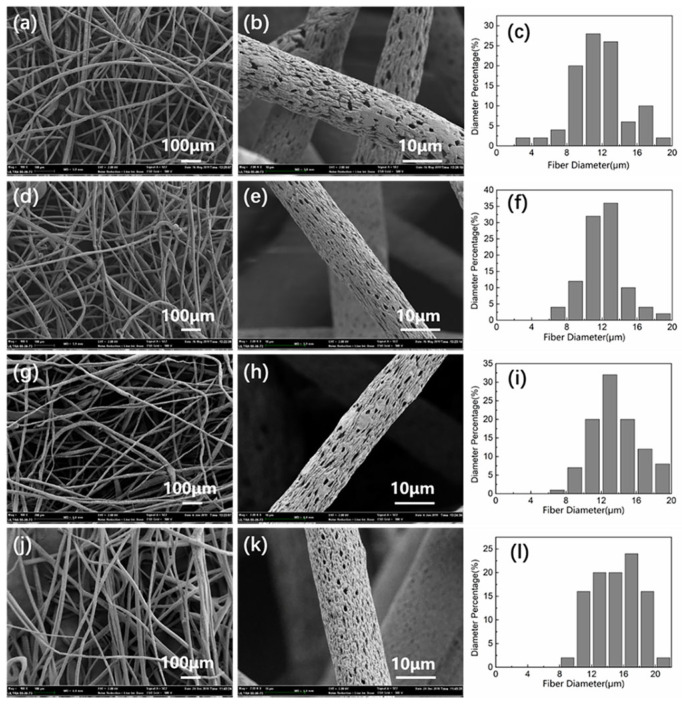
SEM Diagram and Diameter Distribution of Porous Composite Fibers with Different Mass Fractions of MMT: (**a**–**c**) the mass fraction of MMT is 0%; (**d**–**f**) MMT mass fraction is 1%; (**g**–**i**) the mass fraction of MMT is 2%; (**j**–**l**) MMT mass fraction is 3%.

**Figure 2 polymers-14-05458-f002:**
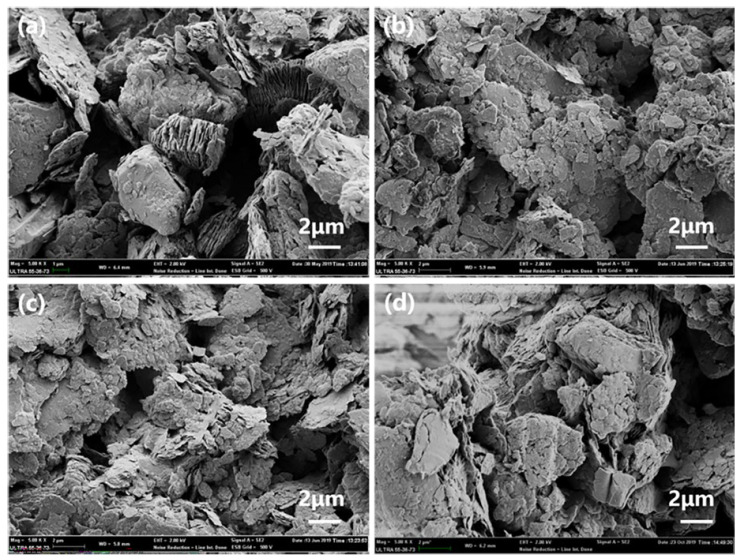
SEM images of various modified MMT: (**a**) MMT (5000×); (**b**) HCl-MMT (5000×); (**c**) SDS-MMT (5000×); (**d**) CTS-MMT (5000×).

**Figure 3 polymers-14-05458-f003:**
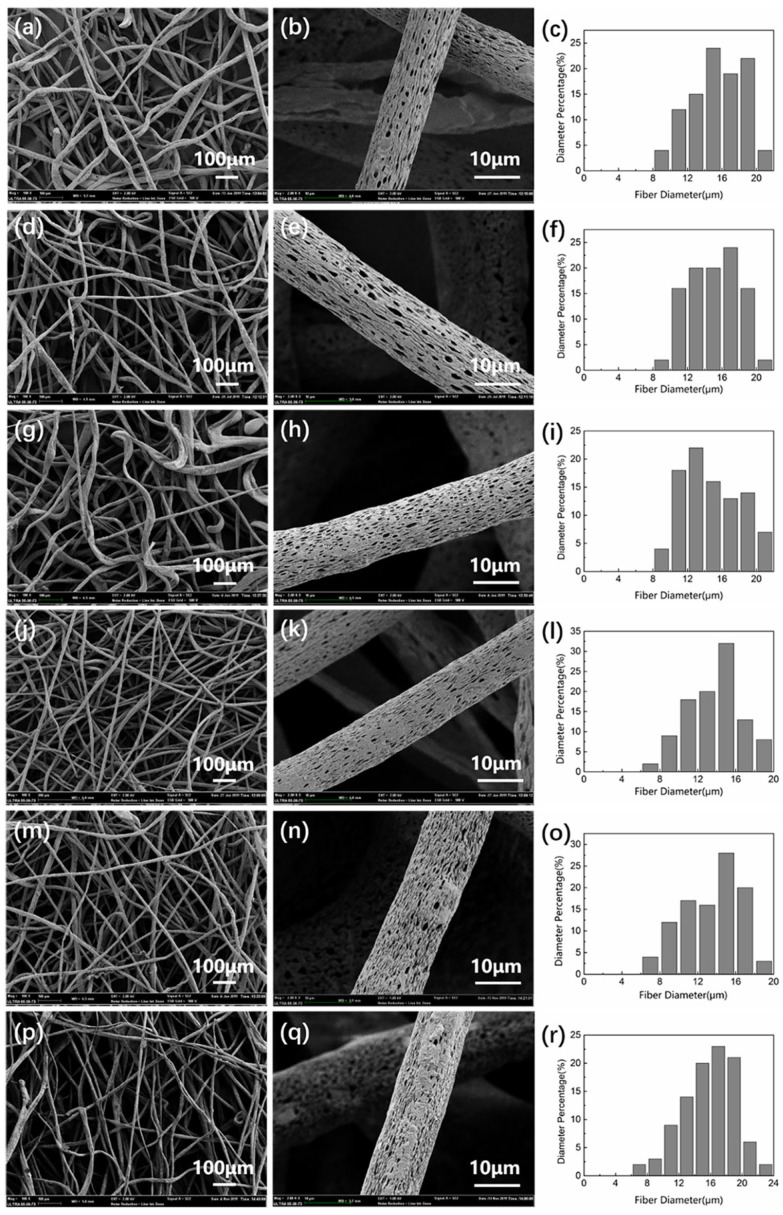
SEM and fiber diameter distribution of porous composite fiber membrane prepared by three kinds of modified MMT with different additions: (**a**–**c**) 3 wt% HCl-MMT/CA; (**d**–**f**) 3 wt% SDS-MMT/CA; (**g**–**i**) 4 wt% SDS-MMT/CA; (**j**–**l**) 3 wt% CTS-MMT/CA; (**m**–**o**) 4 wt% CTS-MMT/CA; (**p**–**r**) 5 wt% CTS-MMT/CA.

**Figure 4 polymers-14-05458-f004:**
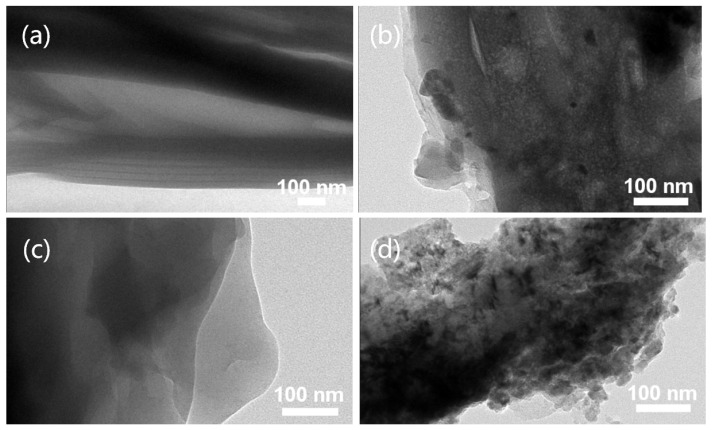
TEM images of porous composite fiber membranes prepared by different modification methods: (**a**) CA; (**b**) 3 wt% HCl-MMT/CA; (**c**) 4 wt% SDS-MMT/CA; (**d**) 5 wt% CTS-MMT/CA.

**Figure 5 polymers-14-05458-f005:**
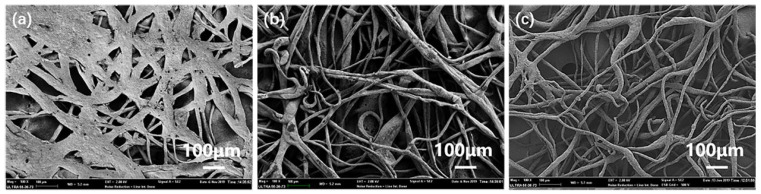
SEM of high-load-modified MMT composite fiber membrane: (**a**) 4 wt% HCl-MMT; (**b**) 5 wt% SDS-MMT; (**c**) 6 wt% CTS-MMT.

**Figure 6 polymers-14-05458-f006:**
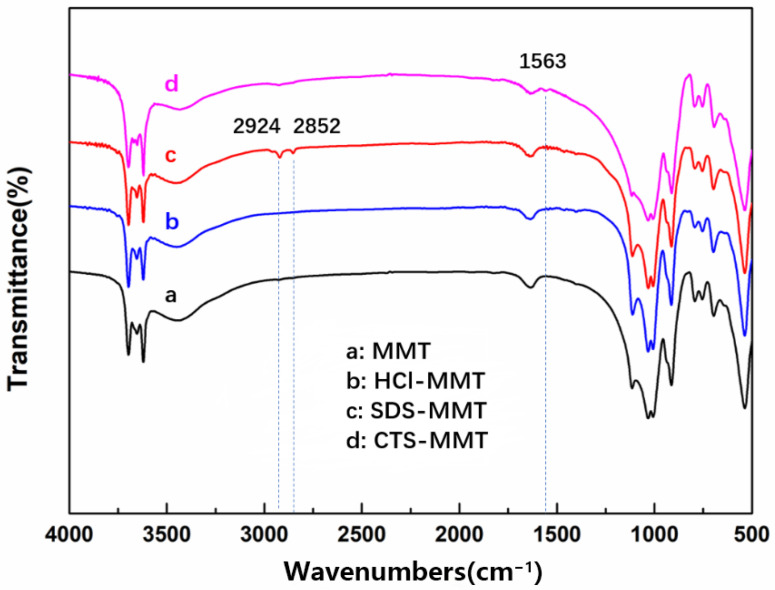
Infrared spectra of MMT and modified MMT.

**Figure 7 polymers-14-05458-f007:**
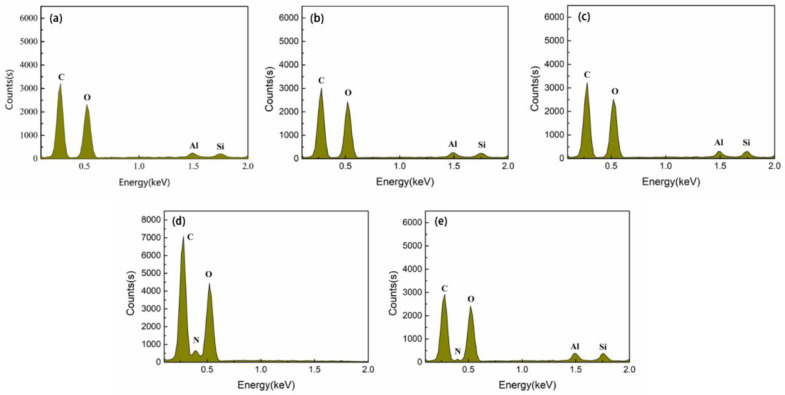
EDS diagram of natural MMT/CA composite fiber membrane modified MMT composite fiber membrane and CTS: (**a**) MMT/CA composite fiber membrane; (**b**) 3 wt% HCl-MMT/CA composite fiber membrane; (**c**) 4 wt% SDS-MMT/CA composite fiber membrane; (**d**) CTS; (**e**) 5 wt% CTS-MMT/CA composite fiber membrane.

**Figure 8 polymers-14-05458-f008:**
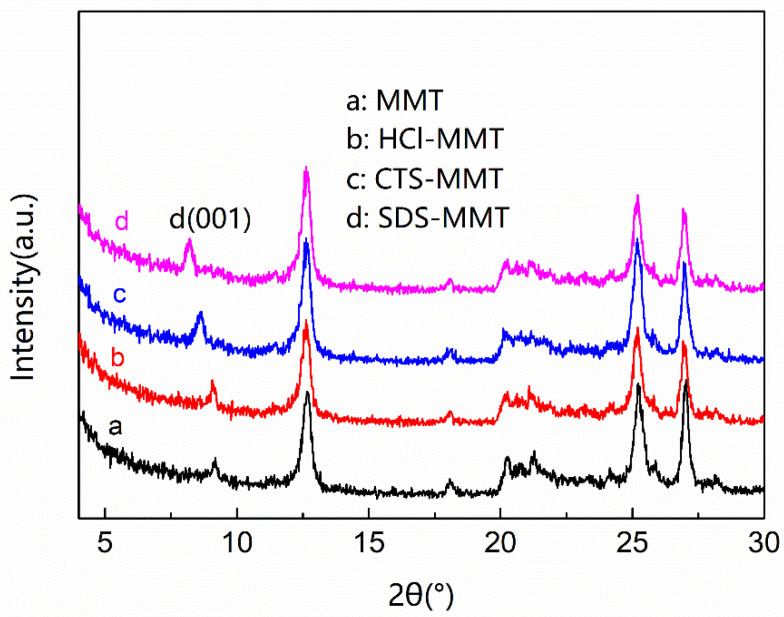
XRD diagram of MMT and modified MMT.

**Figure 9 polymers-14-05458-f009:**
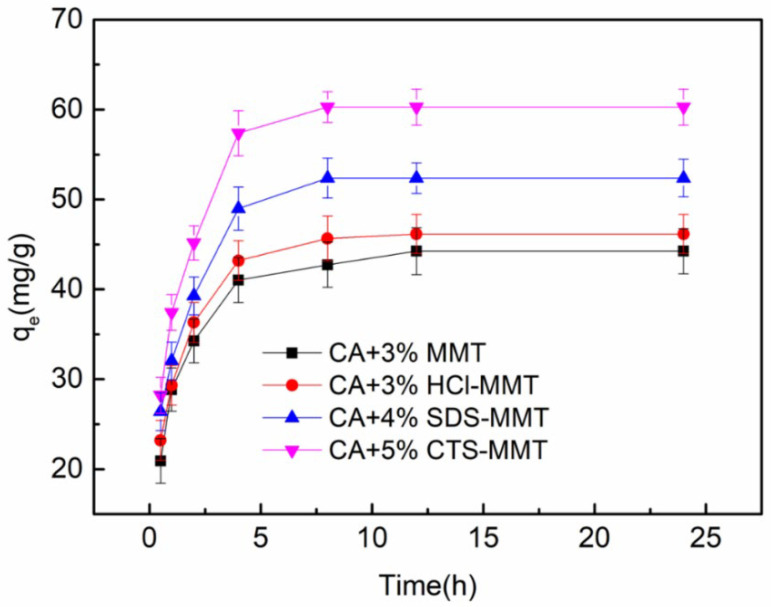
Relationship between adsorption capacity and adsorption time of various composite fiber membranes with pH 5 of Cu^2+^ adsorption solution (100 mg/L).

**Figure 10 polymers-14-05458-f010:**
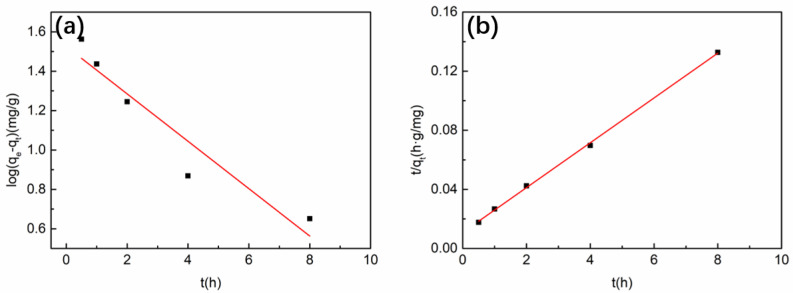
Kinetic fitting diagram of Cu^2+^ (100 mg/L, pH = 5) adsorption on 5 wt% CTS-MMT composite fiber membrane: (**a**) quasi-first-order kinetic model, (**b**) quasi-second-order kinetic model.

**Figure 11 polymers-14-05458-f011:**
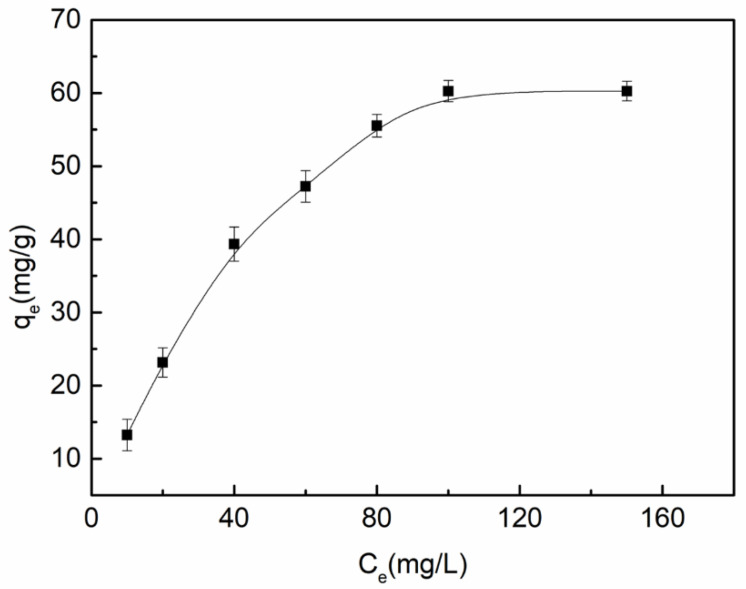
Relation curve between adsorption equilibrium concentration and adsorption capacity of 5 wt% CTS-MMT composite fiber membrane when pH value of Cu^2+^ adsorption solution (100 mg/L) is 5.

**Figure 12 polymers-14-05458-f012:**
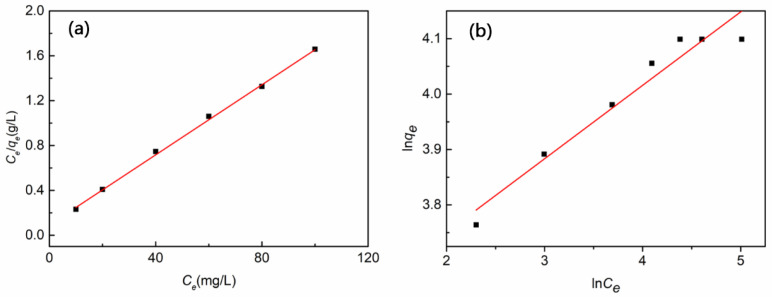
Fitting diagram of adsorption isotherm of Cu^2+^ (100 mg/L, pH = 5) adsorbed by 5 wt% CTS-MMT composite fiber membrane: (**a**) Langmuir model, (**b**) Freundlich model.

**Figure 13 polymers-14-05458-f013:**
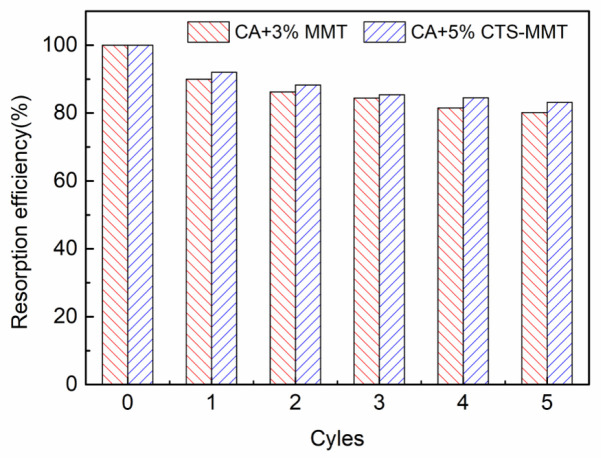
Relation Curve between Reabsorption Efficiency and Cycle Times of 3% MMT and 5% CTS-MMT Composite Fiber Membranes.

**Figure 14 polymers-14-05458-f014:**
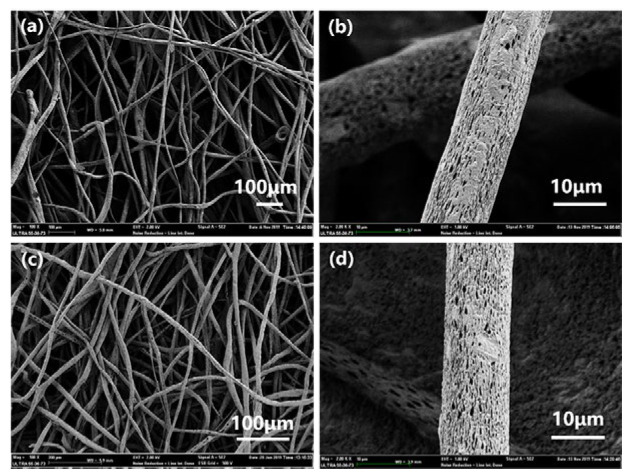
SEM of the first spinning (**a**,**b**) and the second spinning (**c**,**d**) of CTS-MMT composite fiber membrane.

**Table 1 polymers-14-05458-t001:** Kinetic fitting parameters of Cu^2+^ adsorption on 5 wt% CTS-MMT composite fiber membrane.

Pseudo-First-Order Model			Pseudo-Second-Order Model		
*K* _1_	*q_e_*	*R* ^2^	*K* _2_	*q_e_*	*R* ^2^
h^−1^	mg/g		g/mg·h^−1^	mg/g	
0.276	1.525	0.885	0.021	65.92	0.995

**Table 2 polymers-14-05458-t002:** Parameter table of adsorption isothermal model of 5 wt% CTS-MMT composite fiber membrane.

Langmuir Model			Freundlich Model		
*q_m_*	*K_L_*	*R* ^2^	*K_F_*	*n*	*R* ^2^
mg/g	L/mg		L/mg		
64.023	0.072	0.993	1.156	3.354	0.889

## Data Availability

Not applicable.

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
