# Peer review of "Preparation of Copper Ion Adsorbed Modified Montmorillonite/Cellulose Acetate Porous Composite Fiber Membrane by Centrifugal Spinning"

_polymers, 2022, doi:10.3390/polym14245458_

Round 1

Reviewer 1 Report (New Reviewer)

The present manuscript reports on the “Preparation of copper ion adsorbed montmorillonite/cellulose acetate porous composite fiber membrane by centrifugal spinning”. The work is of some interest but seems to be too primitive and lacks novelty, proper scientific support and justification. There are many reports exhibiting adsorption process of copper ions. Thus, in my opinion, the manuscript in its present form cannot be considered for publication.

Following are some of the comments/suggestions which will be useful to the authors.

1. First of all, there are many previous works published for adsorption process. The authors seem deliberately avoid those papers. This is unusual, as the authors need to acknowledge the previous literature and compare their work with the similar ones in the literature and demonstrate their research outcomes in terms of advantages and disadvantages. Some of studies are given below need to cited;

 i- Shahid, M., Farooqi, Z. H., Begum, R., Arif, M., Irfan, A., & Azam, M. (2020). Extraction of cobalt ions from aqueous solution by microgels for in-situ fabrication of cobalt nanoparticles to degrade toxic dyes: A two fold-environmental application. Chemical Physics Letters754, 137645.

ii- Farooqi, Z. H., Irfan, A., & Begum, R. (2021). Gold nanoparticles and polymer microgels: Last five years of their happy and successful marriage. Journal of Molecular Liquids336, 116270.

iii- Arif, M. (2022). Complete life of cobalt nanoparticles loaded into cross-linked organic polymers: a review. RSC Advances12(24), 15447-15460.

iv- Shabir, G., Saeed, A., & Hussain, G. (2019). Synthesis and Optical Study of Sensitive and Selective Calix [4] Based Cu2+ Ion Detection Probes. Russian Journal of General Chemistry, 89(4), 813-818.

v- Arif, M., Shahid, M., Irfan, A., Nisar, J., Wang, X., Batool, N., ... & Begum, R. (2022). Extraction of copper ions from aqueous medium by microgel particles for in-situ fabrication of copper nanoparticles to degrade toxic dyes. Zeitschrift für Physikalische Chemie, 236(9), 1219-1241.

vi- Shahid, M., Farooqi, Z. H., Begum, R., Arif, M., Wu, W., & Irfan, A. (2020). Hybrid microgels for catalytic and photocatalytic removal of nitroarenes and organic dyes from aqueous medium: a review. Critical Reviews in Analytical Chemistry50(6), 513-537.

2. The authors should use complete name in first time appearance. For example, HCl-MMT, SDS-MMT, and CTS-MMT is written by authors in line 18 and 19. Write complete name along with abbreviation.

3. The reference style in the text is not correct. For example, “Anirudhan [14,15] et al. prepared a Polyitaconic acid / methacrylic” in line 45. Write it as “Anirudhan et al. [14,15] prepared a”. Similar mistake is present in throughout the manuscript.

4. The authors write oxidation state of metal ions in two forms. Such as “for Co/ (â…¡) in nuclear power plant wastewater and Th (â…£) in seawater” and “maximum adsorption capacity of Cu2+ on the prepared 5 wt% CTS-MMT composite fiber membrane”. Correct it.

5. The morphology is not clear with SEM results. They should add TEM results in this manuscript.

6.   The authors should cite the different published articles to support their results.

7. Correct “coefficients R2 of the two” from line 409.

8. The reason of better adsorption efficiency of 5% CTS-MMT than others are missing. The reason of these adsorption behaviors should be explained.

9. The present research work should be compared with previous. This increases the importance of this work. 

10. The regeneration of Cu+2 ions after adsorption are missing.

11. the adsorption conditions are missing from Figure 8-11. Write the adsorption conditions such as pH, Temperature, Concentrations etc.

11. Indicate the peaks appeared at 3500 cm-1 in FTIR results.

12. Title of this article should be improved.

Author Response

Reviewer 2 Report (New Reviewer)

This manuscript discussed a new method to design and prepare a new composite fiber membrane to absorb heavy metal ions. The properties of the membrane are measured, and the performance of the absorption is discussed. I have 2 comments on the result and discussion section:

1. The author fitted the data into 2 models, namely the Langmuir and Freundlich adsorption isotherm models. I would like to see more introduction to these two models except the mathematical equations. What is the fundamental difference between these two models physically? And when the data can be described by a certain model, I would like to see some possible reasons or physical explanations for this result. 

2.  When the author ran the fitting to determine whether it is the quasi-first-order or quasi-second-order models, only comparing the R-square is not enough since they are fitted with different X and y. I would like to see some more rigorous hypothesis testing to show which model is correct.

I would not recommend this to be published in Polymers before these comments are addressed properly in the manuscript.

Round 2

Reviewer 1 Report (New Reviewer)

Accept

Reviewer 2 Report (New Reviewer)

My questions are answered properly. Agree to publish on Polymers. 

This manuscript is a resubmission of an earlier submission. The following is a list of the peer review reports and author responses from that submission.

Round 1

Reviewer 1 Report

The manuscript polymers-2023209 "Preparation of montmorillonite/cellulose acetate porous composite fiber membrane for copper ions adsorption by centrifugal spinning" concerns the synthesis of modified montmorillonite/cellulose acetate porous composite fiber as adsorbents of copper ions. Fibers were synthesized using the centrifugal spinning method and modified with HCl, SDS, and CTS.  The authors of the manuscript characterized obtained adsorbents and performed adsorption experiments in different contact times and different initial concentrations of copper ions.

The manuscript presents interesting results, and the research design is appropriate. The manuscript has the potential, however, there are issues that have to be addressed before publication.

General comments:

-        Language needs to be significantly improved by a professional editor. The text is hard to read and not easy to follow, especially at the beginning of the abstract. Extensive editing of English language and style required.

-        Moreover, there are a lot of basic editorial errors and typos throughout the whole text like unnecessary spaces or lack of spaces, unnecessary upper letters, creatures like "HC1" instead of "HCl" etc. Authors must carefully read the manuscript, and all those typos must be corrected. I pointed out some of those errors in the specific comments above, but there are much more of them that should be found and eliminated.

-        There is almost no discussion regarding the preparation of adsorbents. The results are described properly but what arises from them? There is no comparison to the existing literature and no explanation of why centrifugal spinning was chosen and how it affects the results.

-        Why SDS and CTS have been chosen for experiments? Please explain.

-        There is also almost no discussion regarding the adsorption of Cu. Once again, there is only a description of the results with no further discussion. Why do adsorbents reach maximum adsorption at different times and with adsorption capacity? What are the proposed mechanisms of adsorption and how they are different between adsorbents?

-        What was the pH of the Cu solution? Was it measured before and after adsorption? It would give information regarding the mechanisms. I recommend adding the results of adsorption if different pH values.

-        Did the authors try to fit data to the intraparticle diffusion kinetic model? Moreover, state clearly what information regarding adsorption can be obtained from kinetic models. The authors stated that “the adsorption process is more suitable to be explained by the quasi-second-order kinetic model” – what does that mean? What comes from that statement?

-        The results obtained in this study should be compared with those obtained elsewhere in the literature in order to highlight the efficiency of this absorbent.

-        What about the regeneration possibility and reuse of the absorbent?

-        Conclusions should be improved.

Specific comments:

-        Title – Is this a paper about the preparation of adsorbent by centrifugal spinning or adsorption by centrifugal spinning? The title is misleading and has to be rewritten.

-        Line 13-17 – These sentences have a serious language disease, please modify.

-        Keywords should not repeat the ones used in the title.

-        Line 38 – References should be written as [4-7] not [4,5,6,7] – correct that throughout the whole text.

-        Line 43-44 – What is CO(II)? Do you mean Co(II)? Also, remove unnecessary spaces before and in brackets.

-        Line 48 – Remove unnecessary spaces in “mg/g” – correct that throughout the whole text.

-        Line 55 – Correct the typo in reference 21.

-        Line 82-83 – Are you sure that Glacial acetic acid stands as SDS in your paper? In lines 102-108 you refer to SDS as an anionic surfactant (assume that sodium dodecyl sulfate). Glacial acetic acid is not a surfactant.

-        Line 77-85 – Why does each product start with an upper letter since you separate them with a comma? It should be “dimethyl sulfoxide”, “montmorillonite”, “hydrochloric acid” etc.

-        Line 85-88 and 146-154 – Add the details (producer, city, country) of the equipment that was used.

-        Line 91 – Do not use space before % – correct that throughout the whole text.

-        Line 94 – Do not use space before °C – correct that throughout the whole text.

-        Line 98 - HC1?

-        Line 146-154 – Add the details of the analysis e.g., which technique was used for FTIR analysis (DRIFT?), how many scans for 1cm-1? Etc.

-        Line 194 – “Figure 3 shows the SEM and corresponding fiber diameter distribution” – no, Figure 3 does not show the SEM (scanning electron microscope), it shows the SEM images – correct that throughout the whole text.

-        Line 196 – Figure 4 or Figure 3?

-        Line 240 – “Figure 5 shows the FTIR of MMT” – again, no, Figure 4 does not show the FTIR, it shows the FTIR spectrum – correct that throughout the whole text.

-        Line 288-292 – How do those compounds (CTS and SDS) act in the interlayer spaces of montmorillonite? Why is the montmorillonite expanded? Schematic representation and more discussion would be nice.

-        Line 299 – pH not “PH”

-        Table 1 – unify the style

-        Equations - unify the style of numbering

Author Response

请看附件

Reviewer 2 Report

The quailty of this week is good for the Journal. However, language should be improved carefully!!

Reviewer 3 Report

The paper is interesting.

I recommend publication only if the following issues can be addressed.

- The authors must discuss the differences between their work and previous articles.

- Lines 30-34: You should mention that the discharge of heavy metal polluted wastewater (e.g., brine) degrades water quality and thus water cannot be directly used for potable water (via desalination) and industrial applications. Cite the following references:

Panagopoulos, A. (2022). Brine management (saline water & wastewater effluents): Sustainable utilization and resource recovery strategy through Minimal and Zero Liquid Discharge (MLD & ZLD) desalination systems. Chemical Engineering and Processing - Process Intensification, 108944.

Panagopoulos, A., & Giannika, V. (2022). Decarbonized and circular brine management/valorization for water & valuable resource recovery via minimal/zero liquid discharge (MLD/ZLD) strategies. Journal of Environmental Management, 324, 116239.

Panagopoulos, A. (2022). Process simulation and analysis of high-pressure reverse osmosis (HPRO) in the treatment and utilization of desalination brine (saline wastewater). International Journal of Energy Research.

- Much more explanations and interpretations must be added for the Results.

- Conclusion: Include more of your results.

- Conclusion: Discuss the applicability of your findings and future study in this field.

- Language editing is recommended.

- Grammar editing is recommended.

- Add standard deviation to your results/graphs/tables.

- How many replications you performed?

Reviewer 4 Report

REVIEW RESULTS

This manuscript present well-rounded studies reporting the further knowledge about a green and environmentally friendly fibrous composite heavy metal ion adsorption. Three kinds of modified MMT, HCl-MMT, SDS-MMT, and CTS-MMT, were prepared by inorganic modification and organic modification with natural adsorption material montmorillonite (MMT) and cellulose acetate (CA) as the loading substrate, and the porous MMT/CA composite fibre membrane was constructed by centrifugal spinning. The aims of the manuscript and the results of the data are clearly and   concisely stated. The Authors also provided sufficient evidence for the claims they are making. The authors spent a great effort and time to collect the experimental data and to analyse it. The intention (goal) and the approach (method) of this study are acceptable. The literature review also seems to be appropriate. Overall, the idea of this study is very attractive. Accordingly, this paper is considered to be accepted for publication in Polymer Journal MDPI, but requires minor revision.

1.     Please revise the figure of FTIR. For better understanding, the authors are requested to present the existence of the specific peaks observed by FTIR spectrum and the Identification of number of absorption bands in the entire IR spectrum. The analysis results were then compared with the literatures.

2.     The authors are requested to add the information (operating condition) of the equipment that was used for analysis of sample.

3.     The originality of this manuscript still needs to be emphasized especially in introduction part. The authors are requested to describe the differences between the present method with other routes from the previous research.

1.     The index similarity of this paper is average 18%. Please reduce the similarity over range 10%. 

Round 2

Reviewer 1 Report

Almost all comments I had previously, are still legitimate.

1. I do not think that the manuscript has been corrected by a professional editor. Grammar and style still need to be polished.

2. Also, although the authors made some effort in editing and correcting the typos, the manuscript is still full of them. E.g., the authors corrected the typo in line 101 and make another one (lack of space in "HClremaining"). Same in line 46. Authors have to carefully check the whole manuscript, word by word. The present form of editing is not sufficient for scientific publication.

3. There is almost no discussion regarding the adsorbents and adsorption experiments. There is no comparison to the existing

literature and no explanation of why centrifugal spinning was chosen, why is better than other approaches, and how it affects the results. The obtained results are not discussed with existing literature.

Please explain all those concerns in the manuscript, not only in responses to the reviewer.

4. The reason why SDS and CTS have been chosen for experiments should be explained in the manuscript.

5. Why do adsorbents reach maximum adsorption at different times and with adsorption capacity? What are the proposed mechanisms of adsorption and how they are different between adsorbents? All of that should be explained extensively in the manuscript.

6. “It has been proved before the experiment that when pH=5, the porous composite fiber membrane has the best adsorption effect on Cu2+.” - Is that explained in the manuscript?

8. The results obtained in this study should be compared with those obtained elsewhere in the literature in order to highlight the efficiency of this absorbent.

26. It should be explained in the manuscript.

General comment:

I believe that in MDPI requirements the revised manuscript should be prepared in that way, so reviewers can easily see the changes made. All changes that the author made during the revision process should be highlighted or tracking changes in MS Word should be used. Please mind that in the future.

Reviewer 3 Report

The authors improved significantly their manuscript, which is now ready for publication. I recommend acceptance for publication.

Round 3

Reviewer 1 Report

The authors did not make any significant changes during the two rounds of revision. 

I asked for preparing a more extensive discussion of your results, I even proposed a question that you should answer in this discussion. In return, the authors send me quotations from your introduction section. The introduction section is not the place for discussing the results. The discussion still has not been made.  E.g. I recommended adding a small paragraph or a table with a comparison to other adsorbents, it is just a simple quick literature check and writing one paragraph. Mentioning adsorption capacity in the conclusion or introduction is not a comparison of your adsorbent to the other adsorbents used for Cu immobilization. Experiments performed by authors are interesting and could be worth publishing if authors put more effort into putting their results in the existing scientific field, but authors decided to completely ignore all the findings that have been published in the field until now. The authors should place their work in the body of scientific research done by scientists all over the world - this is one of the purposes of discussion. 
